# WHAT MATTERS FOR BATCH ONLINE REINFORCEMENT LEARNING IN ROBOTICS?

**Perry Dong, Suvir Mirchandani, Dorsa Sadigh, Chelsea Finn**
Stanford University

## ABSTRACT

The ability to learn from large batches of autonomously collected data for policy improvement—a paradigm we refer to as *batch online reinforcement learning*—holds the promise of enabling truly scalable robot learning by significantly reducing the need for human effort of data collection while getting benefits from self-improvement. Yet, despite the promise of this paradigm, it remains challenging to achieve due to algorithms not being able to learn effectively from the autonomous data. For example, prior works have applied imitation learning and filtered imitation learning methods to the batch online RL problem, but these algorithms often fail to efficiently improve from the autonomously collected data or converge quickly to a suboptimal point. This raises the question of what matters for effective batch online reinforcement learning in robotics. Motivated by this question, we perform a systematic empirical study of three axes—(i) algorithm class, (ii) policy extraction methods, and (iii) policy expressivity—and analyze how these axes affect performance and scaling with the amount of autonomously collected data. Through our analysis, we make several observations. First, we observe that the use of Q-functions to guide batch online RL significantly improves performance over imitation-based methods. Building on this, we show that an implicit method of policy extraction—via choosing the best action in the distribution of the policy—is necessary over traditional explicit policy extraction methods from offline RL. Next, we show that an expressive policy class is preferred over less expressive policy classes. Based on this analysis, we propose a general recipe for effective batch online RL. We then show a simple addition to the recipe, namely using temporally-correlated noise to obtain more diversity, results in further performance gains. Our recipe obtains significantly better performance and scaling compared to prior methods.[1]

## 1 INTRODUCTION

The success of modern deep learning has hinged on the ability of learning methods to leverage vast amounts of data. In robotics, although recent works have focused on mitigating this gap by proposing large robotic datasets (Open X-Embodiment Collaboration, 2024; Khazatsky et al., 2024), robot learning continues to operate under a substantially smaller data regime than other fields due to the amount of human supervision required for data acquisition. Even in relatively large data regimes, policies based on imitation learning often struggle to achieve reliable performance. Instead of or in addition to manually collecting robotic data, a desirable alternative is to learn policies that can self-improve. Evidence in both robotics and language modeling have suggested that self-improvement via reinforcement learning (RL) can yield substantial performance gains. However, online RL settings as they have been traditionally studied in robotics can be impractical to scale up to deployment scenarios, as they typically involve updating the policy frequently during policy execution to make use of deployment data. Scaling up this paradigm would involve updating large models in the loop of execution and handling potentially unstable or unsafe behaviors as the policy is learning, both of which are challenging practical constraints. Instead, another setting we can consider is to iteratively perform offline improvement with batches of online deployment data. We refer to this as *batch online RL*, a training paradigm in which policies generate large batches of rollouts that are then used to iteratively refine those same policies. This paradigm has the potential to enable truly scalable robot

---

[1]Videos are available at https://pd-perry.github.io/batch-online-rl/

learning by significantly reducing the need for human effort of data collection due to benefits from self-improvement.

Learning from autonomously collected data for policy improvement, however, remains a significant challenge in robot learning as current algorithms struggle to fully leverage this autonomous data (Mirchandani et al., 2024). Prior methods have focused on tackling this through either imitation learning (IL) or filtered-IL methods (Liu et al., 2023; Ahn et al., 2024; Bousmalis et al., 2023). Despite their intuitive appeal, these approaches have yielded suboptimal results. IL methods have inherent limitations in their ability to leverage suboptimal demonstrations within autonomously collected datasets, while methods based on weighted or filtered IL often have diminishing returns and do not scale well with increasing amounts of autonomous data (Mirchandani et al., 2024). This raises the question: *what are the necessary components to enable batch online RL?* We posit that effective use of autonomously collected data requires policies that not only can *collect diverse trajectories* but also *can learn from this diversity*.

In this work, we perform a systematic empirical study to investigate what enables effective batch online RL in robotics with the goal of providing a general recipe to tackle this problem. We break down the key components of batch online RL approaches into several axes—(i) algorithm class, (ii) policy extraction methods, and (iii) policy expressivity—and analyze how each of them affects performance. The first axis we analyze is the algorithm class. Prior approaches to the batch online RL problem in robotics often focus on IL or filtered-IL methods as approaches that are easy to carry out. We compare these to an algorithm class that can more effectively benefit from diverse suboptimal data—value-based RL. We observe that utilizing a Q-function—trained on the cumulatively collected data—to guide the policy enables it to leverage the diversity in the autonomous data to learn new behaviors, thereby overcoming barriers to using autonomous data for self-improvement and scaling significantly better. Among the different methods of using a Q-function, we find that the expressivity of the policy class and policy extraction method are vital choices. Specifically, an expressive policy is necessary to both generate and consume diverse data, and an implicit method of policy extraction, where we choose the best action in the distribution of the policy, significantly outperforms traditional explicit policy extraction methods.

Based on these observations, we propose a general recipe for effective batch online RL: train an expressive IL policy as the actor, train a Q-function on the autonomous data, and perform implicit policy extraction using the Q-function to obtain a policy for autonomous rollouts. On top of the recipe, we propose a simple practical addition to induce even more diversity and achieve better sample efficiency: applying a small amount of temporally correlated noise modeled by the Ornstein–Uhlenbeck process during autonomous rollouts. Overall, our recipe results in up to $2\times$ performance improvement over previous methods on a set of six complex robotic manipulation tasks from Robomimic (Mandlekar et al., 2021), Adroit (Rajeswaran et al., 2018), and MimicGen (Mandlekar et al., 2023). Finally, we validate the practicality of the recipe on a challenging real-world robotics task, improving over the initial policy by 30% in success rate in three iterations of batch online RL.

## 2 RELATED WORK

**Autonomous Improvement.** There has been growing interest in the robotics community in autonomous improvement—where an initial set of demonstrations is used to collect autonomous data which can be leveraged for learning better policies. Several works (Liu et al., 2023; Ahn et al., 2024; Bousmalis et al., 2023; Mirchandani et al., 2024; Zhou et al., 2024) have attempted to use IL or filtered-IL methods as the algorithm for improvement. However, policy improvement can saturate quickly (Mirchandani et al., 2024). This is likely due to the fact that the methods do not effectively leverage suboptimal or failure data, and can lack diversity in the autonomous data that they collect. Nevertheless, these methods can be appealing because of their simplicity and because they avoid some of the practical challenges of running real-world online RL. Zhou et al. (2024) provides a system for autonomous improving policies. However, they operate in a goal-conditioned multi-task setting. Nakamoto et al. (2024) explores improving robot foundation models without training the models themselves, but training a value function for guidance; our recipe outperforms this approach in our real robot experiments.

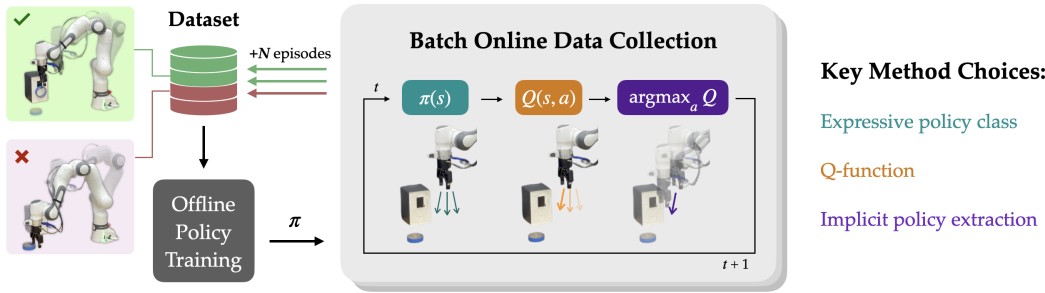

Figure 1: **Overview.** We consider the batch online RL problem setting, in which a policy is trained on an initial dataset, used to collect batches of autonomous data during deployment, and then re-trained on the accumulated dataset. We analyze three critical axes in a spectrum of approaches to the batch online RL problem: policy expressivity, algorithm class, and policy extraction method. We propose a general effective recipe of training an expressive IL policy as the actor, value-based RL to learn a Q-function, and performing implicit policy extraction with the Q-function to get a policy for rollouts.

**Finetuning Offline RL.** While running online RL methods from scratch can be prohibitively expensive, it is possible to accelerate the process by using prior demonstrations. There are several techniques for doing so, such as including the offline data in the replay buffer (Vecerík et al., 2017; Hester et al., 2018; Song et al., 2023; Ball et al., 2023) and regularizing the policy with a behavior cloning loss (Rajeswaran et al., 2018). Additionally, it is possible to use offline RL (Peng et al., 2019; Kumar et al., 2020) to initialize a policy and value function, and then improve these policies with an online fine-tuning procedure (Nakamoto et al., 2023; Nair et al., 2021; Kostrikov et al., 2022; Lyu et al., 2022). However, online fine-tuning can be challenging (Nakamoto et al., 2023; Lee et al., 2021) due to distribution shifts and catastrophic forgetting of the initialization. We operate in a different setting, where large batches of deployment data are collected autonomously using a fixed policy, and then that policy is iteratively refined offline based on the deployment data. This setting has the advantage of decoupling training from data collection during policy deployment.

## 3 PRELIMINARIES

In this section, we describe the batch online RL problem setting, as well as ingredients from prior approaches that are relevant to our empirical study in Section 4.

**Batch Online RL.** Robotics operates under a substantially smaller data regime than other fields due to the difficulty in obtaining data. Instead of or in addition to manually collecting robotic data, a desirable alternative is to enable robots to self-improve. While online RL methods can address this problem in theory, running online RL in the real-world is practically challenging because of the need to train during deployment. The challenges arise from the need to update the policy in the loop of execution, which can be challenging as models scale, and also from the fact that the policy may undergo unstable or unsafe behaviors as the policy is learning. A middle ground is instead to iteratively perform offline self-improvement with batches of online collected data. Various forms of this have appeared in prior works (Matsushima et al., 2020; Riedmiller et al., 2022). We refer to this setting as *batch online RL*.

We formulate batch online RL in the context of a Markov Decision Process (MDP) $\mathcal{M} = (\mathcal{S}, \mathcal{A}, \gamma, p, r, \mu_0)$ (Bellman, 1957). $\mathcal{S}$ denotes the state space and $\mathcal{A}$ denotes the action space. $r(s, a)$ is the reward function mapping from state and action pairs to rewards, $\mu_0$ is a initial state distribution, and $p(s'|s, a)$ denotes the transition dynamics. As in traditional RL, the objective is to find a policy $\pi$ that maximizes the expected sum of discounted rewards $\mathbb{E}_{\tau \sim p^\pi(\tau)}[\sum_t \gamma^t r(s_t, a_t)]$ where $p^\pi(\tau)$ gives the likelihood of a trajectory $\tau$ under $\pi$.

Unlike traditional RL, the policy $\pi$ is frozen during a given deployment. It may, however, be updated offline after each iteration of collecting a batch of data. We provide a general framework for the batch online RL setting in Algorithm 1. An initial policy $\pi_0$ is trained from an offline dataset $\mathcal{D}_0$. Then, for each iteration $i$, the policy $\pi_{i-1}$ is used to collect rollouts $\mathcal{D}_i$, which is appended to the original dataset

and trained on to obtain $\pi_i$. Depending on the specific approach, a $Q$-function may also be trained at each iteration and used to guide rollouts. This is repeated for $N$ iterations to obtain the final policy.

Having introduced the problem setting, we now discuss key ingredients of existing works and problem settings that are relevant to our empirical analysis of batch online RL.

$\mathcal{D}_0 \leftarrow$ Collect offline dataset
$Q_0 \leftarrow \texttt{UpdateValue}(\mathcal{D}_0)$
$\pi_0 \leftarrow \texttt{UpdatePolicy}(\mathcal{D}_0)$
**for** $i$ *in* $1 \ldots N$ **do**
    $\mathcal{D}_i \leftarrow$ Collect $M$ rollouts
       with $\texttt{Rollout}(\pi_{i-1}, Q_{i-1})$
    $Q_i \leftarrow \texttt{UpdateValue}(\cup_i \mathcal{D}_i)$
    $\pi_i \leftarrow \texttt{UpdatePolicy}(\cup_i \mathcal{D}_i)$
**end**

**Algorithm 1:** Framework of Batch Online RL

**Imitation Learning.** Several works (Liu et al., 2023; Ahn et al., 2024; Bousmalis et al., 2023; Mirchandani et al., 2024) have attempted to use imitation learning-based methods as an approach to the batch online RL problem. Imitation learning is often formulated as behavior cloning, which uses supervised learning to learn a policy $\pi_\theta$ parameterized by $\theta$ to maximize the log-likelihood of actions in a dataset $\mathcal{D}, \mathbb{E}_{(s,a)\sim\mathcal{D}}[\log \pi_\theta(a \mid s)]$.
In the batch online RL setting, $\pi_i$ is updated by training on a combination of the base dataset $\mathcal{D}_0$ and its own online rollouts. In the case of *filtered imitation learning*, transition samples are reweighted—with a weighting of 1 assigned to transitions sampled from successful trajectories and a weighting of 0 assigned to failures.

**Value-based RL.** A large body of work in offline RL (Kumar et al., 2020; Kostrikov et al., 2022; Nair et al., 2021; Wu et al., 2019) has studied the problem of learning a policy from a static offline dataset $\mathcal{D}$. As we discuss in Section 4, we consider applying techniques from offline RL within the $\texttt{UpdateValue}$ and $\texttt{UpdatePolicy}$ step at each iteration of batch online RL. We emphasize the distinction from pure offline RL, where no online transitions are collected, and pure online RL, where training occurs during the rollout procedure.

In this work, we primarily consider the Implicit Q-Learning (IQL) (Kostrikov et al., 2022) value objectives given its effectiveness on a range of tasks. IQL aims to fit a value function by estimating expectiles $\tau$ with respect to actions within the support of the data, and then uses the value function to update the Q-function. To do so, it aims to minimize the following objectives for learning a parameterized Q-function $Q_\phi$ (with target Q-function $Q_{\phi'}$) and value function $V_\psi$:

$$L_Q(\phi) = \mathbb{E}_{(s,a,r,s')\sim\mathcal{D}} \left[ \left( r + \gamma V_\theta(s') - Q_\phi(s,a) \right)^2 \right] \tag{1}$$

$$L_V(\psi) = \mathbb{E}_{(s,a)\sim\mathcal{D}} \left[ L_2^\tau \left( Q_{\phi'}(s,a) - V_\psi(s) \right) \right], \tag{2}$$

where $L_2^\tau(x) = |\tau - \mathbf{1}(x < 0)|x^2$. Intuitively, $\tau$ is a hyperparameter that controls how much the value function approaches the maximum of the Q-function, with greater $\tau$ making the value function closer to the maximum. The policy can then be recovered from the Q-function and value function via a *policy extraction* step. As in IQL, we consider Advantage-Weighted Regression (AWR) (Peng et al., 2019) as a canonical example of a policy extraction method. AWR aims to maximize

$$J_\pi(\theta) = \mathbb{E}_{(s,a)\sim\mathcal{D}}[e^{\beta(Q(s,a)-V(s))} \log \pi_\theta(a|s)],$$

where $\beta$ is a hyperparameter to interpolate between behavior cloning and recovering the maximum of the Q-function. In the following section, we consider a spectrum of batch online RL approaches, examining algorithms leveraging IL or value-based RL, different policy extraction methods, and policy expressivity.

## 4 EMPIRICAL ANALYSIS OF BATCH ONLINE RL

We now introduce our analysis setup. Given the general framework of batch online RL as presented in Algorithm 1, we perform a systematic empirical study on a spectrum of approaches to understand key axes that affect performance. Specifically, we analyze the following:

1. **Algorithm class**. We perform experiments with imitation learning and filtered-IL objectives as well as value-based RL objectives for updating the policy (Section 4.1).
2. **Policy extraction method**. We consider two methods of extracting policies from a value-based RL method, which we refer to as explicit and implicit policy extraction (Section 4.2).

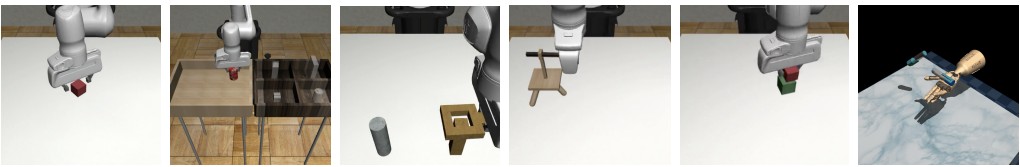

Figure 2: **Simulation environments.** Robomimic tasks: `Lift`, `Can`, `Square`; MimicGen tasks: `Threading`, `Stack`; Adroit tasks: `Pen`.

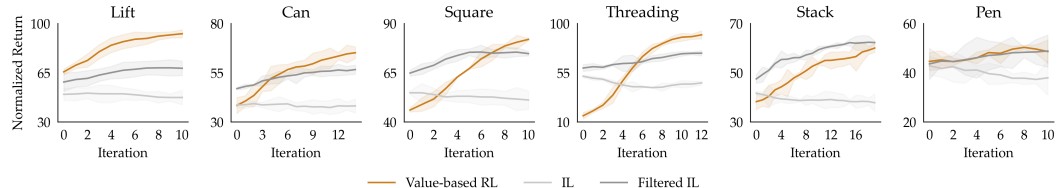

Figure 3: **Normalized returns of different algorithm classes** over multiple iterations of improvement. Value-based RL significantly outperforms IL and filtered-IL. Runs are 3 seeds, 100 evaluations with error bars showing standard error.

3. **Policy expressivity**. We analyze the effects of policy expressivity, focusing on two policy classes: a Gaussian policy and an expressive diffusion-based policy (Section 4.3).

**Experimental Setup.** We perform our study in simulation and validate our findings on a real-robot manipulation task. We choose six challenging continuous control environments in simulation: `Lift`, `Can`, and `Square` from Robomimic (Mandlekar et al., 2021); `Stack` and `Threading` from Mimicgen (Mandlekar et al., 2023); and `Pen` from Adroit (Rajeswaran et al., 2018). We illustrate these environments in Figure 2. The size of $\mathcal{D}_0$ varies from 5 to 100 demonstrations depending on the task difficulty; we choose this size such that the base policy $\pi_0$ performs at a success rate between 30–65%, putting it in a realistic scenario that leaves room for improvement. We run $N$=10 to 20 iterations of batch online RL with $M$=200 rollouts per iteration.

Based on our results, in Section 5 we present a recipe for batch online RL, and demonstrate the practicality of the recipe on a challenging real-world robotic task of hanging tape on a hook.

## 4.1 WHICH ALGORITHM CLASS WORKS BEST?

We first perform a controlled set of experiments to identify the extent to which algorithm class affects performance in batch online RL. Three classes of algorithms are commonly used in prior works; (**a**) **IL**, which is the most straightforward approach for learning from autonomous rollouts, (**b**) **filtered-IL**, which filters low-quality rollout trajectories prior to re-training to ensure that the policy only taps into successful demonstrations, and (**c**) **value-based RL** which attempts to additionally learn from negative data via learning a value function and performing a Bellman update.

For all of the algorithm classes, we use a diffusion-based policy as the default. For value-based RL, the default extraction method we consider is only using the Q-function for guidance; specifically, during rollouts, multiple actions $\{a_i\}_{i=1,2,...,N}$ are sampled from the policy and the best action $\arg\max_{a_i} Q(s, a_i)$ is selected. Since we use a diffusion policy as the base policy, this can be instantiated as Implicit Diffusion Q-Learning (IDQL) (Hansen-Estruch et al., 2023). We examine policy extraction choices individually in more detail in Section 4.2 and policy expressivity in Section 4.3.

**Value-based RL is necessary for overcoming suboptimal convergence of Filtered-IL.** In Figure 3, we present the average normalized returns over iterations of batch online RL for each algorithm class on our six tasks. We observe that value-based RL methods tend to significantly outperform IL-based methods. Vanilla IL performs the worst on all tasks, which is perhaps not surprising as vanilla IL will fit the failure trajectories of the autonomous rollouts. Filtered-IL, while exhibiting an initial improvement, often converges quickly to suboptimal performance compared to value-based RL.

**Value-based RL generates and consumes more diverse data than imitation-based methods.** We hypothesize that the better performance of value-based RL methods is a result of a better ability to leverage diversity in autonomously collected data. We examine heatmaps for state visitation of trajectories during batch online RL in Figure 4. We see that value-based RL methods result in much more diverse trajectories after batch online RL. Intuitively, this makes sense because value-based RL methods can use the Q-function to determine which states and actions are desirable even in failure trajectories, thus allowing the policy to learn from a more diverse set of trajectories including failures.

**Value-based RL scales better with larger batches of autonomous data.** An important desiderata for choosing an algorithm class for the batch online RL problem setting is the ability to scale performance with the amount of collected at each iteration. For each environment, we set $M$ to a small, medium, and large value: 50, 100, and 200 trajectories for Robomimic and Mimicgen, and 100, 200, and 300 for Adroit Pen. We present the results of data scaling in Figure 5. We see that value-based RL achieves the best scaling across the board. In all but one task, value-based RL performs

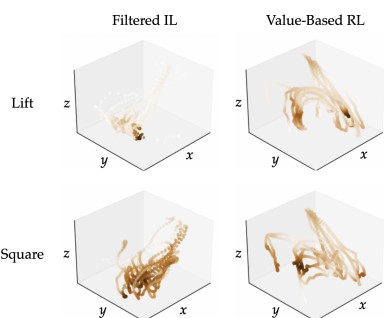

Figure 4: **Heatmap of state visitations of successful trajectories** after batch online RL for value-based RL and filtered IL on Lift and Square. 3D plots of end-effector positions are shown for each task. Darker colors correspond to higher density of visitation. Note that value-based RL methods achieve more state diversity in successful rollouts.

significantly better as the amount data increases, suggesting stronger ability to leverage large batches of data for improvement. This is in contrast to IL or filtered-IL which tend to saturate with more data collected at each iteration.

**Value-based RL is necessary but not sufficient for batch online RL.** One takeaway from this section is that for batch online RL, we cannot get away with just doing IL or filtered-IL as many prior works suggest. But if this is the case, why have prior works seen limited benefits of value-based RL in practice (Mirchandani et al., 2024; Mandlekar et al., 2021)? We posit that value-based RL as the algorithm class for updating the policy is *necessary but not sufficient*. There are other choices that are critical for value-based RL methods to work for batch online RL—specifically, policy extraction method (Section 4.2) and policy class (Section 4.3).

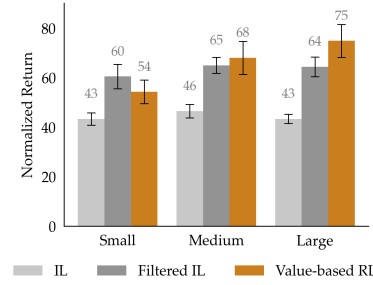

Figure 5: **Normalized returns** of different algorithm classes at various data scales averaged across all tasks.

## 4.2 HOW TO EXTRACT THE POLICY?

Given the advantages of value-based RL compared to IL and filtered-IL methods in the batch online RL setting from Section 4.1, the second axis we consider is how to extract the policy from the value function. We separate policy extraction into two distinct categories, explicit policy extraction and implicit policy extraction, to analyze the effect of extraction method on performance.

(*a*) **Explicit policy extraction.** Prior works in offline RL typically carry out policy extraction following the idea of maximizing the Q-value while staying close to the behavior policy. This is done through maximizing a RL objective and an IL objective offline to learn the policy. This approach has the advantage of explicitly learning on signals from the Q-function, while still making the policy stay close to the behavior dataset. For our experiments, we select Advantage-Weighted Regression (AWR) (Peng et al., 2019) as a canonical example of explicit policy extraction that follows this principle, as described in Section 3.

(*b*) **Implicit policy extraction.** In contrast to a separate policy extraction step that explicitly extracts a policy to maximize Q-values, an alternative for policy extraction is to optimize for the best actions in the distribution of the policy, which we refer to as implicit policy extraction. One simple approach for carrying this out is to select actions online by querying the Q-function, where multiple actions are sampled from the policy and the highest Q-value action is selected. While implicit policy

Figure 6: **Normalized returns of explicit versus implicit policy extraction.** *Pre* refers to the initial base policy $\pi_0$ trained on $\mathcal{D}_0$ and *Post* refers to the policy after completing iterations of batch online RL. Returns are averaged over 3 seeds and 100 evaluation trials at each iteration with error bars showing standard error.

extraction loses potentially useful signals from the Q-function for the policy, it has the advantage of disentangling the value function and policy training, which provides more stable learning.

**Implicit policy extraction significantly outperforms explicit policy extraction in batch online RL.** Figure 6 shows the average normalized returns before (*Pre*) and after (*Post*) running batch online RL with explicit versus implicit policy extraction. The number of iterations varies for each task; refer to Figure 3 for details on iterations. Interestingly, we find that although explicit policy extraction achieves a stronger *initial* performance in nearly every benchmark task, implicit policy extraction performs significantly better after running batch online RL. In fact, explicit policy extraction does not improve the policy performance on any of the tasks. This result indicates that the policy extraction approach is critical in batch online RL, likely due to implicit policy extraction leveraging diverse autonomous data more effectively: while explicit policy extraction approaches provide more signals from the value function for policy learning, this signal becomes detrimental when new, diverse data is added during batch online RL and causes a shift in action distribution. The policy extracted from explicit policy extraction cannot adjust to this shift as well as implicit policy extraction, resulting in subpar performance.

## 4.3 DOES THE EXPRESSIVITY OF THE POLICY MATTER?

From the preceding analysis, we observe that value-based RL outperforms IL and filtered-IL in batch online RL settings. One natural question is whether value-based RL with implicit policy extraction is sufficient, or whether the choice of policy class also matters for performance. The third axis we analyze is expressivity of the policy class. We focus on two classes: a Gaussian policy and an expressive diffusion-based policy. Both use supervised IL objectives and the value function objectives from IQL.

(*a*) **Gaussian policy.** Gaussian policies model the mean and variance of $\pi(a|s)$ and sample actions from the learned mean during rollouts. Though they are a less expressive class of policies, Gaussian policy are still worth examining because they are fast for inference, which is especially desirable in real-world tasks. On top of that, offline RL methods such as ReBRAC (Tarasov et al., 2023) based on Gaussian policies have shown performance on par to diffusion based methods, and current online RL methods based on Gaussian policies have been successful even in challenging real-world tasks (He et al., 2025; Lin et al., 2024; Yin et al., 2024; Lin et al., 2025), suggesting perhaps less expressive policies can be just as effective for value-based RL.

(*b*) **Expressive diffusion policy.** Diffusion-based policies are a highly expressive policy class which use a Markovian noising and denoising process to model the behavior distribution of the data. This allows them to better model multimodal action distributions. A further advantage of diffusion-based policies is that they lend themselves well to implicit policy extraction approaches as expressive policy classes can capture a more diverse distribution more effectively.

**Expressive diffusion policies with implicit policy extraction outperform Gaussian policies with explicit policy extraction.** In Figure 7, we compare the performance of expressive policies with Gaussian policies before and after running multiple iterations of batch online RL. For the expressive policy class, we use implicit policy extraction as analyzed in Section 4.2. For Gaussian policies, since the action distribution is less expressive, we use explicit policy extraction. We find that across all tasks and environments, the former significantly outperforms the latter.

**Expressive policies enable better implicit policy extraction.** To control for the advantages of the implicit policy extraction method in batch online RL that we observed in Section 4.2, we additionally run a version of the Gaussian policy with implicit policy extraction by sampling actions from the

Figure 7: **Normalized returns of value-based RL with diffusion versus Gaussian policy** before and after improvement. To address confounding of policy extraction methods, we show both explicit and implicit policy extraction approaches for Gaussian policies. Returns are averaged over 3 seeds and 100 evaluation trials at each iteration with error bars showing standard error.

learned mean and variance and using the Q-function for guidance during rollout. Although there is a significant improvement over Gaussian policies with explicit policy extraction, the overall performance is considerably worse than that of an expressive policy. With the expressive policy, the initial policy can better capture the action distribution which ultimately leads to stronger improvement during batch online RL. Why might less expressive Gaussian policies be sufficient for online RL but not batch online RL? In online RL, the action distribution $\pi(a|s)$ modeled by the policy is always changing as the policy is updated each step. This means there will always be new actions taken, so the policy does not need to have a good model of the action distribution from the initial dataset to enable improvement. This is in contrast to batch online RL, where to leverage diversity of the online data, the initial model needs to have captured enough of an expert action distribution to collect useful trajectories that can then be used to improve the policy.

## 5 RECIPE FOR BATCH ONLINE RL

Based on our analysis from Section 4, we propose the following recipe for batch online RL: train an expressive IL policy as the actor, train a Q-function on the autonomous data, and perform implicit policy extraction using the Q-function to obtain a policy to do rollouts. We illustrate the full recipe in Figure 1. In our experiments, we instantiate the recipe with a diffusion-based policy network trained with IL, a Q-function trained via the IQL objective, and implicit policy extraction through sampling actions and choosing the one with the highest Q-value during rollouts. This instantiation recovers the IDQL algorithm (Hansen-Estruch et al., 2023) for one iteration of batch online RL, though the recipe defines a category of methods and can be instantiated with other expressive policy classes, implicit policy extraction methods, and Q-function objectives.

**Improving diversity with temporally-correlated noise.** The components in each axis share a common theme of better leveraging diverse data in autonomous collection. We apply a simple addition to our recipe to induce more diversity. The idea is to add a small amount of temporally correlated noise, modeled as an Ornstein-Uhlenbeck process, to the policy actions during the autonomous rollouts. This idea has been successful in online RL previously (Lillicrap et al., 2016) and extends to the batch online RL setting as well. We present the results of returns averaged over all tasks in Figure 8 and refer to Section 4.1 for a detailed setup of scaling experiments. We find that the addition of temporally-correlated noise enables higher performance at each data level. However, it does not improve data scaling because the correlated noise has the effect of increasing the distribution the policy learns, but this increase in distribution can be naturally achieved with more data. This suggests that temporally-correlated noise can be a valuable addition, though our recipe does not hinge on it.

**Real-world robotic manipulation task.** To validate the practicality of our proposed recipe, we conduct an experiment with running batch online RL on a challenging real-world vision-based robotic manipulation task. The task involves controlling a 7-DoF Franka Research 3 robot to grasp a roll of tape and hang it onto a hook. We visualize the initial and final states of the task in Figure 9. We use RGB images and robot proprioceptive state (joint and end-effector positions) as input, and use a ResNet-18 (He et al., 2015) as the vision backbone. We use one wrist camera and one camera mounted on as an external view to stream the RGB images.

We collect 5 initial demonstrations in $\mathcal{D}_0$ and run $N = 3$ iterations of batch online RL, each with $M = 30$ rollouts. We compare our recipe with filtered-IL and a steering baseline adapted from (Nakamoto et al., 2024), where we train the Q-function on $M = 90$ rollouts (as well as $\mathcal{D}_0$)

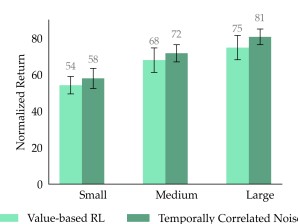

Figure 8: **Normalized returns** of value-based RL with and without temporally correlated noise at different data scales, averaged over all tasks.

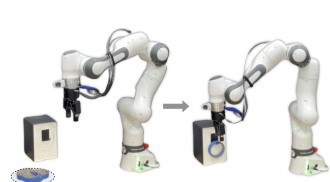

Figure 9: **Real-world task** of hanging tape on a hook. The shaded blue area (left) depicts the tape's initial distribution. The right side shows successful task completion.

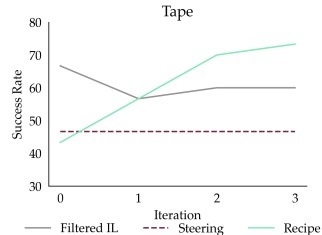

Figure 10: **Results for real-world Tape task** over iterations of batch online RL. Our recipe outperforms filtered-IL and steering with a Q-function.

corresponding to the same amount of data our recipe trains on for the last iteration, and use a policy trained on $\mathcal{D}_0$ to sample actions. Our recipe obtains 30% improvement over the initial policy in just 3 iterations (Figure 10). Filtered-IL does not improve upon the initial performance, likely because the policy was able to capture the action distribution of the $\mathcal{D}_0$ fully in pre-training. The steering baseline performs the worst, indicating that it is necessary to train the policy on the new rollouts.

## 6 DISCUSSION

As robotic models become ever more capable, it becomes increasingly important to find ways to improve models beyond simply collecting more data. Batch online RL provides a paradigm for just that—enabling policies to leverage their own rollouts for self-improvement without the complications of online RL. For practitioners, our analysis offers a clear, practical recipe for executing batch online RL. For researchers, we bring to attention open questions for future work to optimize each component of the recipe further. For example, are techniques from offline RL the most optimal for learning a Q-function, or do we need something beyond pessimism specific to batch online RL? In the same vein, can we improve implicit policy extraction beyond choosing the actions with the best Q-value? We believe solving these questions will result in significantly better and more capable self-improving robotic models. We will open source the code for the final recipe.

## 7 LIMITATIONS

In this work, we empirically analyze the key axes that affect performance in batch online RL, demonstrating that the general recipe of value-based RL, implicit policy extraction, and an expressive policy class enables effective self-improvement of policies in this setting. Our work presents a general recipe on batch online RL, though it does have a number of limitations. First, we focus on robotic tasks with a continuous action space for the study. In problems that have a discrete action space or if we apply a discretization scheme to the actions, the results might not directly transfer as the Q-function can exhibit different properties, policy extraction is different as the Q-function can be used as a policy, and the notion of an expressive and non-expressive policy class is not well defined. We propose adding temporally-correlated noise in the rollouts for better sample efficiency. However, directly adding noise may not be applicable in some deployment settings, though we find empirically that adding a small amount of noise only changes the success rate of the policy marginally. Lastly, in our experiments the initial policy achieves some level of success. How to better start from a completely non-successful policy for batch online RL is an interesting direction for future work.

## 8 ACKNOWLEDGMENTS

This work is in part supported by AFOSR YIP, ONR grant N00014-22-1-2293, NSF #1941722, and the NSF CAREER award. Chelsea Finn is a CIFAR fellow.

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

# APPENDIX

## A  ADDITIONAL EXPERIMENTS

**Data scaling.** In this section, we present additional scaling experiments for the scaling portion of Section 4.1 and the temporally correlated noise portion of Section 5. We present the performance of the methods for each task instead of an average over the tasks.

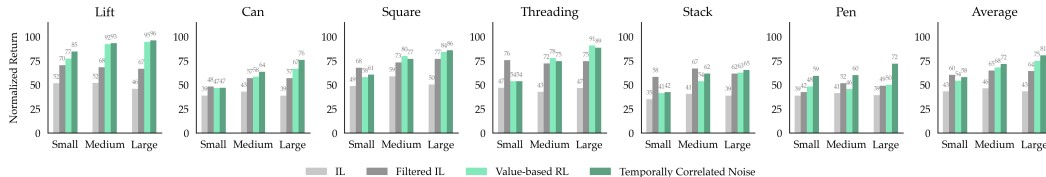

Figure 11: **Normalized returns** of value-based RL compared with IL, filtered-IL, and temporally-correlated noise at different data scales, shown for each task.

From Figure 11, we see that value-based RL scales better with data in nearly every task, while IL-based methods either do not scale or scale in a limited capacity. In addition, temporally-correlated noise outperforms not adding temporally correlated noise for each task and data scale. Temporally correlated noise is especially useful for Adroit Pen, which has been known in the literature to benefit from more exploration.

**Value-based RL with more iterations for Square and Stack.**  Because of compute restrictions, the results reported in the main paper for Robomimic Square and MimicGen Stack were converged for IL-based methods but not for value-based RL. We report the results for value-based RL run for a longer number of iterations in Figure 12. We see that the difference between IL methods and value-based RL becomes larger with more iterations.

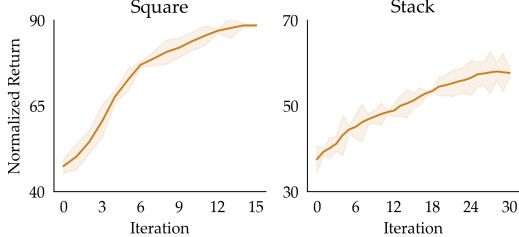

Figure 12: **Normalized returns of value-based RL** for Robomimic Square and MimicGen Stack. Error bars show standard error over 3 seeds.

**Value-based RL with more rollouts per iteration.**  We also report runs for value-based RL with more rollouts per iterations for environments that saturated prematurely. From Figure 13, we see that value-based RL often exceeds premature saturation just with more data in each iteration.

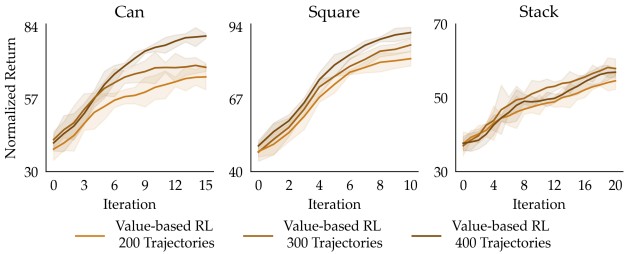

Figure 13: **Normalized returns of value-based RL** for more rollouts per iteration for Robomimic Can, Square and MimicGen Stack. Error bars show standard error over 3 seeds.

# B EXPERIMENT DETAILS

**Training Parameters.** We set the number of iterations from $N=10$ to $20$ depending on the environment and $M=200$ rollouts per iteration. We choose the number of trajectories in the initial dataset such that the base IL policy can get 30 to 65% normalized returns prior to batch online RL. For temporally correlated Ornstein-Uhlenbeck (OU) noise, we select one $\theta$ and $\sigma$ value for each environment suite. For implementation, we use the same residual block structure for the policy as IDQL (Hansen-Estruch et al., 2023) for both expressive policy RL and imitation learning. We use a simple MLP for Gaussian policies.

| Tasks | Parameters | Values |
|---|---|---|
| Robomimic Lift | Dataset Size | 5 |
|  | OU $\theta$ | 5 |
|  | OU $\sigma$ | 0.05 |
| Robomimic Can | Dataset Size | 10 |
|  | OU $\theta$ | 5 |
|  | OU $\sigma$ | 0.05 |
| Robomimic Square | Dataset Size | 100 |
|  | OU $\theta$ | 5 |
|  | OU $\sigma$ | 0.05 |
| MimicGen Stack | Dataset Size | 20 |
|  | OU $\theta$ | 5 |
|  | OU $\sigma$ | 0.03 |
| MimicGen Threading | Dataset Size | 50 |
|  | OU $\theta$ | 5 |
|  | OU $\sigma$ | 0.03 |
| Adroit Pen | Dataset Size | 3 |
|  | OU $\theta$ | 0.1 |
|  | OU $\sigma$ | 0.03 |
| All Tasks | Batch Size | 256 |
|  | Learning Rate | 3e-4 |
|  | IQL Expectile | 0.8 |
|  | Discount | 0.99 |
|  | Number of Sampled Actions | 64 |
|  | Optimizer | Adam |
|  | Beta Schedule | Variance Preserving |
|  | Diffusion Steps | 100 |
|  | Diffusion Policy: MLP Hidden Dim | 256 |
|  | Diffusion Policy: Num Residual Blocks | 3 |
|  | Gaussian Policy: MLP Hidden Dim | 256 |
|  | Gaussian Policy: MLP Hidden Layers | 3 |

Table 1: Hyperparameters for each simulation task. The values specified under All Tasks are shared for different tasks.

**Data Sources.** For each task, the dataset consists of expert trajectories. In Robomimic tasks, we use the Proficient Human dataset provided by Mandlekar et al. (2021). In MimicGen environments, we use the dataset provided by the benchmark (Mandlekar et al., 2023). For Adroit, we use the dataset from D4RL (Fu et al., 2020).

**Evaluation Protocol.** Evaluations are performed by rolling out the policy from start states randomly sampled from the default initial state distribution of the task. The rollout length for `Lift`, `Can`, and `Square` is 400; for `Stack` is 200; for `Threading` is 400; and for `Pen` is 100. Results in the main text report normalized return averaged over 3 seeds and 100 evaluation trials each.

## C    REAL-WORLD TASK DETAILS

In this section, we provide more information on the real world Tape task in our analysis.

**Setup Description.** The Tape task involves hanging a roll of tape onto a rack by controlling a 7-DoF Franka Research 3 robot. To successfully complete the task, the robot needs to precisely aim for and grasp the roll of tape and hang it to the hook. The initial distribution is a roughly 15 cm × 15 cm area. We illustrate an example initial state, success state, and the initial state distribution in Figure 14. The RL agent sends actions to the robot at 5Hz with a maximum episode length of 200 timesteps. The robot obtains visual RGB input from two Intel RealSense D435 cameras, one on the mounted on the end effector and one mounted on the side.

| Sample Initial State | Success State | Initial State Distribution |
| --- | --- | --- |

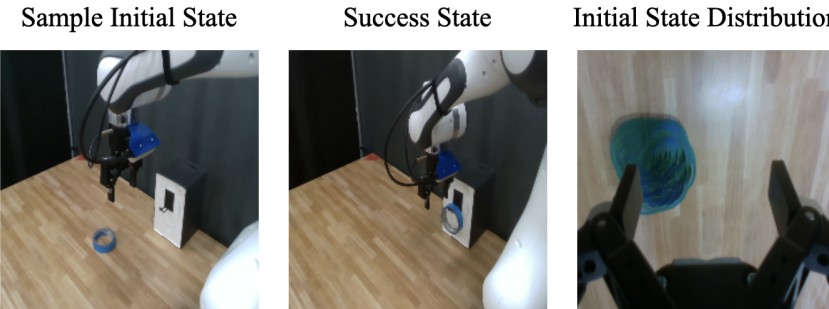

Figure 14: Scenes showing sample initial and success state and the initial state distribution of the real-world Tape task.

**Success Detection.** The Tape task contains a success state that must be reached for the rollout to be considered successful, namely having the tape on the rack. We use a scripted rule to detect if this state has been reached and if there is a success. For each environment step, we utilize a color threshold to check the color of the pixel above the hook. We manually select the pixel location and verify the error of the success detection is near zero.

**Resets.** We perform automatic resets of the Tape environment in our experiments. For a successful rollout, we replay a pre-recorded trajectory to grasp the tape and lift it off the hook. For a failed rollout, we detect the location of the tape and execute a primitive to lift the tape. In both cases, after lifting the tape, we sample an initial state from the initial state distribution and place the tape at the initial state location for the next episode.

