# OpenReview forum: "What Matters for Batch Online Reinforcement Learning in Robotics?"
_ICLR.cc/2026/Conference — ICLR 2026 Poster_

### Official Review · Reviewer_AUz8 · 2025-10-29

**Soundness:** 3
**Presentation:** 3
**Contribution:** 3
**Rating:** 6
**Confidence:** 3

**Summary:**

The paper studies batch online RL setting for robot manipulation, where a fixed policy collects large batches of autonomous rollouts, then the policy (and optionally a Q-function) is retrained offline before the next deployment. The authors systematically vary three axes: (i) algorithm class (imitation learning or filtered imitation learning versus value-based RL), (ii) policy extraction (explicit versus implicit), and (iii) policy expressivity (Gaussian versus diffusion). They conclude that value-based RL with an expressive policy and implicit action selection, implemented by sampling multiple actions from the policy and choosing the argmax under the learned Q, works best, and that adding small Ornstein-Uhlenbeck noise during rollout yields further gains. Experiments on six simulated manipulation tasks and a real robot tape-on-hook task show strong improvements and better scaling trends than imitation-learning baselines.

**Strengths:**

The paper is well structured, easy to follow, and clearly motivated. Problem setting, assumptions, and goals are stated upfront, which makes the experimental choices and takeaways straightforward to understand.

The work decomposes batch online RL into three axes (algorithm class, policy extraction, and policy expressivity) and systematically compares design choices within each axis. This comparative study is valuable to the community because it clarifies which ingredients matter in practice, turns ad hoc intuitions into evidence-backed guidance, and provides an actionable recipe that reliably beats imitation learning baselines.

**Weaknesses:**

1. Limited novelty: The paper’s primary contribution is a careful, systematic comparison of existing ingredients rather than a new objective or algorithm. The proposed recipe largely recombines known components and matches IDQL when run for a single iteration, so the conceptual contribution feels incremental even though the comparative insights are useful.

2. Unclear hyperparameter tuning and fairness: The paper does not clearly explain how hyperparameters for different ingredients were selected, what ranges were explored, or whether tuning effort and budgets were balanced across methods and tasks. This makes it hard to attribute gains to the method rather than choices like K (implicit samples), β or expectile τ, diffusion steps, model capacity, or critic settings. I would suggest documenting the tuning protocol (search space, selection criteria, per-task vs global tuning, seeds)

**Questions:**

See Weaknesses

---

> ### Author Response · Authors · 2025-12-03
> **Official Comment by Authors**
>
> Dear Reviewer AUz8,
>
>
> Thank you for your feedback!
>
> We appreciate your comments that the paper is “well structured, easy to follow, and clearly motivated” with a comparative study that is “valuable to the community” and an “actionable recipe.”
>
> We would like to address your concerns below.
>
> >  The paper’s primary contribution is a careful, systematic comparison of existing ingredients rather than a new objective or algorithm.
>
> The main contribution of our work is an empirical analysis of the key axes of batch online RL, and as a result of this analysis, we propose a general recipe for batch online RL. We are not focused on one specific algorithm, but rather on understanding the design choices that are consequential for algorithms in this setting to be effective. Given these components, we certainly believe there are algorithmic improvements that can be made within this recipe, and we leave it for future work to build on it and make further improvements.
>
>
> > Documenting hyperparameter tuning
>
> Thanks for pointing this out. In practice, we set the number of implicit samples K to 64 and diffusion steps to 100 as default values without tuning. We also set the diffusion model capacity to the same as IDQL, and only tune three values of expectile from [0.7, 0.8, 0.9] for value-based methods following the conventional IQL tuning procedure. We will make sure to document the tuning protocol in the final version.

---

### Official Review · Reviewer_MA26 · 2025-10-29

**Soundness:** 2
**Presentation:** 3
**Contribution:** 2
**Rating:** 4
**Confidence:** 4

**Summary:**

This paper empirically studies what matters for batch online reinforcement learning for robotic tasks. In batch online RL, where agent training and data collection are decoupled, the agent alternates between online data collection by rolling out the current policy and offline policy improvement trained on all data collected so far. The authors hypothesize that an effective batch online RL method should be able to collect diverse data throughout the process. They should also be effective in learning from the diverse dataset collected. The authors examine three key components of the method design: (1) IL-based or Value-based RL (IQL), (2) explicit (AWR) or implicit policy extraction, and (3) expressive (Diffusion-based) or non-expressive (Gaussian) policy. The authors confirm that expressive value-based RL with implicit policy extraction is a feasible recipe for batch online RL in robotics, which achieves both high performance and scaling ability on six simulation tasks and one real-world task.

**Strengths:**

- The paper is well-structured and easy to read.
- The problem setting, batch online RL, is practical and promising. And it makes sense to decompose the problem into the proposed 3 components.

**Weaknesses:**

__There are missing experimental details I would like to verify further:__

- For experiments in Section 4.1, my understanding is: all three methods train a policy with the DDPM objective. The value-based RL one trains an additional Q function guiding policy rollout only. If this is the case, why does the initial performance of value-based RL differ from the IL baseline when both of them learn from $\mathcal{D}_0$?
- In Section 4.2, what is the training objective when the authors apply AWR to a diffusion-based policy with the DDPM objective? Did the authors reweigh the MSE by the advantages, like how IDQL does in their paper (Appendix F.5)?
- In terms of the real-world task, when applying the Steering baseline, why did the authors not train the Q function in a batch online RL manner? I am aware of the Steering paper’s setting and how it is deployed in this paper, but a reasonable variant should consider iteratively improving the Q function on new online rollouts (e.g., N=3, M=30). This also benefits the understanding of whether the performance bottleneck results from the value learning for online batch RL.

__Some arguments made in the paper were not well-supported:__

- In my opinion, the central claim of this paper is: “policies that not only can collect diverse trajectories but also can learn from this diversity matter for batch online RL” (Lines 064-066). Intuitively, I agree that expressive RL policies with value guidance should satisfy this requirement and thus work well for batch online RL. However, the normalized score is an indirect metric supporting this claim. Like Figure 4, I expected to see more direct quantitative analysis or visualization of the diversity of the collected data in each iteration, along the three proposed axes. For example, although an implicit Gaussian policy performs worse than the recipe, showing it might be able to collect relatively more diverse data for Threading and Square tasks can actually enhance the central claim.
- The argument made in Lines 346-351 is not supported with empirical evidence. Could the authors confirm whether implicit policy extraction can generate and leverage more diverse data? Could the authors confirm that explicit policy extraction cannot adapt policy to value functions trained on new data? Is the value learning the performance bottleneck, or the policy learning? Is the data collection the bottleneck?
- In Lines 395-402, could the authors provide direct evidence showing that an expressive policy can capture multimodality in action distribution in the initial iteration? How does the ability of modelling action distribution change across the iterations?

__Despite applying existing methods to a new problem setup, I felt the novelty of this paper is limited.__

- First of all, although I agree that diverse data collection and learning from diverse data are essential for batch online RL, I think the community has been aware of their necessity for offline RL, imitation learning, and online RL. Early prior study have shown their importance in each setup, like [1, 2, 3, 4]. Given that the batch online RL setup could be treated as consecutive offline RL or “low update-to-data ratio” online RL, what are the other unique insights the reader may grasp?
- For the second aspect of policy extraction, I recall that IDQL found that explicit policy extraction via AWR does not bring benefit to the DDPM-trained diffusion policy.

__Minor Issues:__

- In equation (1), the V function should be parameterized by $\psi$.
- In Line 292, it should be “the amount of data collected”.
- In Line 261, the notation of N is abused.
- In Lines 147-149, the authors argue that online RL risks of unsafe and unstable behavior. I wonder if in batch online RL, the agent also risks this since it is possible to conduct unsafe and unstable behavior during the data collection stage if the offline training was not sufficient. But I agree that this should happen less than the online RL paradigm.
- Compared with the network used for diffusion-based policy, the network used for the Gaussian policy in Section 4.3 is smaller (3-layer MLP). Although I do not expect that increasing the model size for the Gaussian policy can significantly improve performance, the authors should mention in the main body this experimental detail.


__References:__

[1] Don't Change the Algorithm, Change the Data: Exploratory Data for Offline Reinforcement Learning

[2] Behavior Transformers: Cloning k modes with one stone

[3] Synthetic Experience Replay

[4] The Primacy Bias in Deep Reinforcement Learning

**Questions:**

Please refer to the questions above __"Minor Issues"__ part.

---

> ### Author Response · Authors · 2025-12-03
> **Official Comment by Authors**
>
> Dear Reviewer MA26,
>
> Thank you for your feedback on our work. We would like to address your questions and concerns below.
>
>
> > Why does the initial performance of value-based RL differ from the IL baseline when both of them learn a DDPM policy?
>
> The difference in performance is because value-based RL uses a Q-function to guide inference while IL directly samples from the policy. The Q-function trained on the offline data can help or hurt performance depending on how well it estimates the values, which is dependent on the offline dataset among other factors.
>
>
> > What is the training objective when the authors apply AWR to a diffusion-based policy with the DDPM objective?
>
> We modify AWR to a diffusion-based policy by reweighting the DDPM objective with advantages, as in the IDQL paper.
>
>
> > Why did the authors not train the Q function for the steering baseline in a batch online RL manner?
>
> To clarify, the Q-function for the steering baseline is indeed trained on the online data. We will make this clearer in the paper. The difference is the policy is not updated with the online data, which shows that steering is not enough and we need to update both the value function and policy with online data.
>
>
> > Visualization of the diversity of the collected data in each iteration, along the three proposed axes
>
> Thank you for the suggestion! We will add a visualization of the collected data in each iteration along the three axes to the final version.
>
>
> > Could the authors confirm that explicit policy extraction cannot adapt policy to value functions trained on new data?
>
> We show in Figure 6 that explicit policy extraction generally does not improve in policy performance after iterations of batch online RL. The value function in this case is trained on new data, but despite that the policy is not able to improve.
>
>
> > In Lines 395-402, could the authors provide direct evidence showing that an expressive policy can capture multimodality in action distribution in the initial iteration? How does the ability of modelling action distribution change across the iterations?
>
> In the first iteration of Gaussian Implicit versus Diffusion Implicit (Figure 7, “Pre”), the difference in performance comes from how well the policy models the action distribution, which is determined by the expressivity. Across iterations, we see both Gaussian and Diffusion policies improve in performance, but Diffusion policy improves much more than Gaussian policies because of the expressivity of the policy.
>
>
> > I wonder if in batch online RL, the agent also risks this since it is possible to conduct unsafe and unstable behavior during the data collection stage if the offline training was not sufficient. But I agree that this should happen less than the online RL paradigm.
>
> As you note, we would expect this to happen less than the online RL paradigm, where the policy gets updated frequently during the rollout procedure. A major advantage of the batch online RL setting is that the policy is fixed during each deployment, which mitigates some of the risks of unstable training that could happen with online updates.
>
> > Notations and increasing the model size for the Gaussian policy can significantly improve performance
>
> Thanks for pointing this out! We will address these notation problems in the final paper. We will also mention in the main body this experimental detail regarding model size.

---

### Official Review · Reviewer_7bd7 · 2025-11-01

**Soundness:** 3
**Presentation:** 3
**Contribution:** 1
**Rating:** 2
**Confidence:** 4

**Summary:**

This paper investigates the problem of "batch online reinforcement learning," a paradigm where a robotic policy is improved by collecting large batches of autonomous data and then performing offline updates. The authors conduct a systematic empirical study to determine the key components for effective learning in this setting. They analyze performance across three axes: (i) algorithm class (imitation learning vs. value-based RL), (ii) policy extraction methods (explicit vs. implicit), and (iii) policy expressivity (Gaussian vs. diffusion policies). The main findings suggest that value-based methods significantly outperform imitation-based ones, implicit policy extraction is more effective than explicit extraction, and expressive diffusion policies are superior to less expressive Gaussian ones. Based on these results, the paper proposes a general "recipe" for effective batch online RL in robotics.

**Strengths:**

*   The paper addresses a practical and important problem. The "batch online RL" setting, which involves collecting large batches of data for offline updates, is a sensible and scalable approach for real-world robotics, reducing the need for constant human supervision and avoiding the instability of purely online updates.
*   The work is structured as a controlled study that ablates different components of the learning pipeline (algorithm, extraction, expressivity). This systematic approach helps to isolate the factors that contribute most to performance.

**Weaknesses:**

*   A major weakness is the lack of motivation for choosing the three specific axes of analysis. The paper repeatedly refers to these axes but never explains why these are the most critical or representative components to study. The selection of only three algorithm classes (IL, filtered-IL, and value-based RL) also feels restrictive and lacks justification, especially when hybrid methods exist.
*   Several of the key findings seem intuitive or are well-established principles in reinforcement learning. For instance, the conclusion that value-based methods lead to more diverse data and better performance than simple imitation learning is not a surprising result. The paper would be stronger if it more clearly situated these findings in the context of what is already known and highlighted what is genuinely new.
*   The paper frames its final contribution as a general "recipe" for batch online RL, but the supporting experiments are conducted on a handful of simulation tasks and a single real-world task. This may not be sufficient evidence to support such a broad claim. Presenting the findings as a set of recommendations or case studies might be a more accurate and defensible framing. The claim that value-based RL is "necessary but not sufficient" is also very strong and may not be fully supported by the evidence.
*   There appears to be a contradiction between the claims made in the text and the data presented in the figures. For example, the paper states that "we cannot get away with just doing IL or filtered-IL," but the results in Figure 5 suggest that these methods do show performance gains and scale with more data, even if they saturate earlier than value-based methods.

**Questions:**

*   Could the authors provide a stronger justification for why these three specific axes—algorithm class, policy extraction, and policy expressivity—were chosen as the focus of the investigation?
*   The introduction states that current algorithms "struggle to fully leverage" autonomous data. Could you elaborate on the specific failure modes of prior works that motivate this study?
*   Why was the comparison of algorithm classes limited to IL, filtered-IL, and value-based RL? There are other approaches, such as those that interpolate between an IL and RL loss, that could have served as relevant baselines.
*   What was the rationale for using a diffusion-based policy as the default across all experiments?
*   For the filtered-IL baseline, what was the specific threshold or criterion used to filter "low-quality" trajectories?
*   In the experimental setup, the initial datasets are chosen to yield a base policy with a 30-65% success rate. How can we be sure that the conclusions drawn from this "realistic scenario" generalize to other starting conditions (e.g., starting from a much worse or better policy)?
*   Regarding Figure 3, the plot lines are difficult to distinguish. Could the visualization be improved for clarity? It would also be helpful for the figure to be self-contained by explicitly mentioning that the value-based method shown is IDQL.

---

> ### Author Response · Authors · 2025-12-03
> **Official Comment by Authors**
>
> Dear Reviewer 7bd7,
>
> Thanks for your thoughtful feedback. We would like to address your questions and concerns below.
>
>
> > Could the authors provide a stronger justification for why these three specific axes—algorithm class, policy extraction, and policy expressivity—were chosen as the focus of the investigation?
>
> From our initial explorations, we found that these three axes were particularly consequential for performance, which is why we chose these three to structure the investigation for the final paper. The paper does explore other aspects like size of the offline dataset, which is presented in the Appendix.
>
>
> > Could you elaborate on the specific failure modes for prior works struggling to fully leverage autonomous data?
>
> Multiple prior works in robotics (e.g. https://arxiv.org/pdf/2211.08416, https://arxiv.org/pdf/2401.12963, https://arxiv.org/pdf/2411.01813) have attempted to use filtered imitation learning to learn from deployment data. However, this method can struggle with low rates of improvement and suboptimal converged performance, as supported by our experimental results. Indeed, imitation-based methods struggle to fully leverage autonomous data, as they ignore useful signals from failure trajectories, and also do not benefit from state and action pairs that can eventually result in successful completions that need to be learned by stitching.
>
>
> > Other approaches such as those that interpolate between an IL and RL loss that could have served as relevant baselines
>
>
> The focus of our work is not testing how good specific algorithms are, but rather what are the key choices that enable batch online RL. To this end, we study the key axes and within these axes, analyze how varying components in these axes affect performance.
>
>
> > What was the rationale for using a diffusion-based policy as the default across all experiments?
>
> We use a diffusion-based policy as the backbone as evidence in both imitation learning and reinforcement learning have shown more expressive policies result in better performance. As we show in Figure 7, the conclusions apply to Gaussian policies as well, albeit with lower overall performance compared to diffusion policies because diffusion policies can better capture complex distributions.
>
>
> > For the filtered-IL baseline, what was the specific threshold or criterion used to filter "low-quality" trajectories?
>
> The trajectories for filtered IL are filtered based on whether they achieve success. The environments have a binary indication of whether the trajectories are successful upon completion, which is used to filter.
>
>
> > Several of the key findings seem intuitive or are well-established principles.
>
> While some of our findings are intuitive, we would argue that certain findings in this work go against previously established conventions, and provide insight for the design of batch online RL algorithms. For example, in purely online RL, using Gaussian policies is the well-established standard, and this paper shows using Gaussian policies lead to poorer performance in the batch online setting. As another example, most offline RL methods rely on explicit policy extraction where the value function is used as gradient signal for the policy update, but we show that in the batch online RL setting, implicit extraction tends to perform better. Regarding our comparative study between different algorithm classes, reviewer jz8H also notes that “some findings are quite interesting, such as value-based methods outperforming the filtered IL after several iterations of batch training but not initially.”
>
>
> > Could the Figure 3 visualization be improved for clarity?
>
> Thank you for the suggestion. We will increase the contrast of the plot lines in the final version.

---

### Official Review · Reviewer_JZ8h · 2025-11-03

**Soundness:** 3
**Presentation:** 3
**Contribution:** 3
**Rating:** 6
**Confidence:** 3

**Summary:**

This paper investigates algorithmic choices in the batch online RL setting, where a limited number of training and data collection iterations are interleaved. Three aspects are evaluated including value-based methods vs. imitation learning, policy extraction methods and the expressivity of the policy. These are done on simulated robotics environments and a real-world task.

**Strengths:**

The topic---batch online RL---is relevant to current applications of RL.
The paper is well-written. The use of color for denoting the sections is a nice touch.
This paper does a good job at exploring some important algorithmic considerations for the batch online RL setting and the findings are presented clearly.

Some findings are quite interesting, such as value-based methods outperforming the filtered IL after several iterations of batch training but not initially.

**Weaknesses:**

My main concern is the lack of certain baselines in the experimental sections.

- What are the previous baseline methods for the benchmarks considered? While I see mulitple comparisons in each section to examine the impact of various choices, I do not see any explicit comparison to prior methods. While achieving the best performance is not strictly necessary, it would be helpful to have some baseline numbers to get a sense of how well the evaluated algorithm is doing.

- In Fig.7, it seems like a natural baseline woudl be to also include the Diffusion policy with the explicit policy updates. Is there a particular reason this was not included?
This seems like a major reason why there is more diversity with the value-based method, which is presented as an advantage (line 282). It's not clear that Fig.4 is a fair comparison if the filtered-IL method used a Gaussian policy instead.

**Questions:**

- Fig.4 is unclear. It is not entirely obvious that there is greater diversity with the value-based method, particularly on the "Square" task. Perhaps it would be better to add a numerical measure of the diversity of states.
Keeping either the 2d or 3d plots alone might be more clear also.


- Is the temporally-extended noise that important? Some papers have shown that independent Gaussian noise can be just as effective [1,2].

- How does the value-based approach compare to some model-based methods? Line 285 posits that value-based methods outperform the policy-based ones due to utilizing bad trajectories more effectively. This property would seemingly be shared by model-based approaches.

[1] CleanRL implementation of DDPG.

[2] "Addressing function approximation error in actor-critic methods" Fujimoto et al.

---

> ### Author Response · Authors · 2025-12-03
> **Official Comment by Authors**
>
> Dear Reviewer JZ8h,
>
> Thank you for your insightful feedback on our work. We would like to address your concerns below.
>
>
> > In Fig.7, it seems like a natural baseline would be to also include the Diffusion policy with the explicit policy updates. Is there a particular reason this was not included? … It's not clear that Fig.4 is a fair comparison if the filtered-IL method used a Gaussian policy instead.
>
> In Figure 4, the imitation learning baseline is indeed a diffusion policy. The difference is the imitation learning baselines do not estimate a value function for the updates whereas value-based RL estimates a value function. For the comparisons in Figure 7, while we do not include an explicit policy extraction for diffusion policies, we note that this comparison is made in Figure 6 where we specifically analyze policy extraction approaches.
>
>
> > Numerical measure of the diversity of states
>
> Thank you for the suggestion! We will add a numerical measure of diversity to the final version of the paper to supplement the plot. Per your suggestion, we will additionally make the 2D and 3D plot more clear to better indicate the diversity of the states.
>
>
> > Is the temporally-extended noise that important?
>
> In our initial experiments, we found that independent Gaussian noise does not achieve the same effect as temporally extended noise in the batch online RL setting. We will include results on independent Gaussian noise in the Appendix of the final version.
>
>
> > How does the value-based approach compare to some model-based methods?
>
> Our work focuses on model-free methods and mainly compares IL based methods and value based RL methods. We agree that it is possible for model-based methods to leverage suboptimal trajectories in this setting, and we leave this question for future work.

---

### Meta-Review · Area_Chair_kydF · 2026-01-07

**Summary:**

**Summary of the paper**

This paper presents a systematic empirical study of batch online RL for robotics, where policy training alternates between collecting large batches of rollouts with a fixed policy and performing offline updates on the accumulated dataset. The authors analyze three design axes: (i) algorithm class (IL / filtered-IL vs value-based RL), (ii) policy extraction (explicit vs implicit), and (iii) policy expressivity (Gaussian vs diffusion). Across six simulated manipulation tasks and one real-world tape-on-hook task, the study reports that value-based RL scales better than IL-based methods with more autonomous data, implicit policy extraction improves over explicit extraction after multiple batch iterations, and expressive diffusion policies outperform Gaussian policies in this setting. Based on these findings, the paper proposes a “recipe” combining an expressive IL actor, a learned Q-function, and implicit policy extraction, with an additional practical tweak of temporally correlated noise to increase rollout diversity.

**summary of the reviewers' concerns**

Reviewer 7bd7 provided a negative assessment (rating 2) and argued the paper’s motivation and contribution framing are weak, particularly questioning why the three chosen axes are the “most critical,” and whether the key findings are truly new versus intuitive. Reviewer  also expressed concern that the paper’s “recipe” claim is too broad given experiments on a limited set of tasks, and flagged an apparent tension between text and figures, noting that IL and filtered-IL still improve and scale with data in some plots even if they saturate earlier.

Reviewer JZ8h (rating 6) was generally positive about the relevance and clarity, but raised missing baselines and fairness concerns: they asked for clearer baseline performance relative to prior work on the same benchmarks, and questioned whether certain comparisons are fair. Reviewer also asked about the importance of temporally extended noise vs i.i.d. Gaussian noise and suggested model-based comparisons as a potentially relevant point of reference.

Reviewer MA26 (rating 4) raised concerns about missing experimental details (e.g., objectives used when applying AWR to diffusion policies, details of the steering baseline training, why value-based differs initially from IL if both use DDPM), and argued that some central claims (about diversity collection/consumption) are not directly supported by quantitative diversity evidence. Reviewer also questioned novelty, noting that related communities already recognize the importance of diversity, and pointed out several minor clarity issues.

Reviewer AUz8 (rating 6) was positive about the clarity and value of the systematic study, but also flagged limited novelty (largely recombining known ingredients) and asked for clearer documentation of hyperparameter tuning and fairness across methods.

**Reviewer Concerns:**

The rebuttal substantively addressed many of the concerns that appear decision-critical.
1. In response to Reviewer JZ8h, the authors clarified that  “explicit policy updates for diffusion” are analyzed in a separate extraction-focused comparison. They also committed to adding a numerical diversity metric to complement Fig. 4, and stated they will include results comparing OU noise vs i.i.d. Gaussian noise in the final version. The model-based comparison was acknowledged as out-of-scope for the paper’s focus on model-free methods.

2. In response to Reviewer 7bd7, the authors justified the choice of the three axes as those that were most consequential in initial explorations, and clarified the failure modes of filtered IL (slow improvement and suboptimal convergence; inability to leverage failures and “stitching”). They also defended using diffusion as the backbone to test the “expressivity” axis, and clarified that filtered-IL filtering uses binary success at trajectory completion. They also agreed to improve figure clarity.

3. In response to Reviewer MA26, the authors answered the missing-details questions: initial performance differs because value-based uses Q-guided inference, AWR for diffusion is implemented by advantage-reweighting the DDPM objective (as in IDQL), and the steering baseline’s Q-function is trained on online data but the policy is not updated. They also committed to adding a per-iteration visualization of collected data diversity and to fixing notation/model-size disclosure issues.

4.  In response to Reviewer AUz8, the authors provided a concrete tuning protocol (e.g., default K and diffusion steps, matching IDQL capacity, and tuning expectile over a small set), and committed to documenting it clearly.

**Reviewer Scores:**

All reviewers are very likely to increase their score.

---

### Decision · Program_Chairs · 2026-01-26

Accept (Poster)